



# Consistent increase of East Asian Summer Monsoon rainfall and its variability under climate change over China in 34 coupled climate models

Anja Katzenberger[1,2] and Anders Levermann[1,2,3]

[1]Potsdam Institute for Climate Impact Research, Potsdam, Germany
[2]Potsdam University, Potsdam, Germany
[3]LDEO, Columbia University, New York, USA

**Correspondence:** Anja Katzenberger (anja.katzenberger@pik-potsdam.de)

**Abstract.** The East Asia Monsoon (EAM) dominates the climate over the densely populated East China and adjacent regions
and is therefore influencing a fifth of the world's population. Thus, it is highly relevant to assess the changes of the central
characteristics of the East Asian Summer Monsoon (EASM) under future warming in the latest generation of coupled climate
models of the Coupled Model Intercomparison Project Phase 6 (CMIP6). Using 34 CMIP6 models we show that all models
that capture the EASM in the reference period 1995-2014 within two standard deviations project an increase in June-August
rainfall independent of the underlying emission scenario. The multi-model mean increase is 17.2% under SSP5-8.5, 12.7%
under SSP3-7.0, 11.9% under SSP2-4.5 and 11.2% under SSP1-2.6 in the period 2081-2100 compared to 1995-2014. For
China, the projected monsoon increase is slightly higher (12.1% under SSP1-2.6 and 19.1% under SSP5-8.5). The EASM
rainfall will particularly intensify in South-East China, Taiwan as well as North Korea. The multi-model mean indicates a
linear relationship of the EASM rainfall depending on the global mean temperature relatively independent of the underlying
scenario: Per degree of global warming, the rainfall is projected to increase by 0.14mm/day which refers to 3% of rainfall in
the reference period. It is thus predominately showing a "wet-region-get-wetter" pattern. The interannual variability is also
robustly projected to increase between 7.0% under SSP1-2.6 and 31.4% under SSP5-8.5 in the multi-model mean between
2050-2100 and 1965-2015. Comparing the same periods, extremely wet seasons are projected to occur 6.5-times more often
under SSP5-8.5.
**Keywords:** East Asian Monsoon, Monsoon, CMIP6, climate models, China

## 1 Introduction

The climate over East Asia is dominated by the monsoon seasons which are defined as reversing seasonal winds between the
Pacific Ocean and the East Asian continent associated with different rainfall regimes. Rainfall during the East Asian Summer
Monsoon (EASM) accounts for 40–50% of the annual precipitation in South China and 60–70% of the annual precipitation in
North China (Lei et al., 2011) making it a central factor for the socioeconomic livelihoods in the region.





During mid may, rainfall surges over the South China Sea establishing a planetary-scale monsoon rainband extending from
the South Asian marginal seas to subtropical western North Pacific. The monsoon then gradually progresses towards inland
resulting in the synchronized onset of the Indian monsoon season as well as the the monsoon season in China and Japan in
early June (Wang et al., 2002). During the summer months, low level southerly winds transport moisture to East China, Korea
and Japan where it converges within the rain belt that is called the Meiyu in China, the Baiu in Japan, and the Changma in
Korea. The wind direction follows the pressure gradient resulting from a zonal land-sea thermal contrast varying throughout
the course of a year (Ha et al., 2012; Wang et al., 2002). The rainfall reaches its maximum in late June over the Meiyu/Baiu and
in late July over northern China. Then, the rainy season retreats progressively poleward in East Asia during July and August,
while southward in the Indian summer monsoon (Wang et al., 2002).
Since the East Asian Monsoon is located in the subtropics - unlike other monsoon systems, it is additionally influenced by
mid-latitude disturbances and convective activity (Ha et al., 2012). Besides, the EAM interacts with various climatological pat-
terns on various time scales, including El Niño-Southern Oscillation (ENSO), the Arctic Oscillation (AO), the Indian summer
monsoon, spring Eurasian snow cover and the thermal forcing of the Tibetan Plateau (Ha et al., 2012).
The progressing and retreat of the Meiyu belt is associated with a large variability of precipitation over East Asia and
accompanied by floods and droughts with potentially devastating impacts on socioeconomic livelihood (Yihui et al., 2020).
In June and July 2020 large parts of East and South Asia were flooded as a result of excessive monsoon rainfall affecting
approx. 35 mio. individuals (Volonté et al., 2021). Therefore, assessing the climate model projections of the East Asian summer
monsoon under climate change is of critical importance for national and regional management strategies.
The central approach to assess changes in the East Asian monsoon throughout the 21st century are global climate models.
The general circulation models (GCM) participating in the Coupled Model Intercomparison Project (CMIP) have provided
some insight regarding future changes of the EAM. The models from the previous generation (CMIP5) project an increase of
the East Asian Monsoon of 10–15% throughout the 21st century under RCP6.0, most pronounced over the Baiu region and
over the north and northeast of the Korean Peninsula (Seo et al., 2013). The strengthening of monsoon rainfall is attributed to an
increase in evaporation as well as moist flux convergence induced by the (north) westward shift of the North Pacific subtropical
high (Lee and Wang, 2014; Seo et al., 2013). Besides, the $CO_2$-induced strengthening of the land-sea thermal contrast plays a
central role for the Asian monsoon (Endo et al., 2018). Chen and Sun (2013) find that the frequency and intensity of intense
precipitation events are also projected to significantly increase over East Asia under RCP4.5.
The continuous development of the GCMs in CMIP has also lead to the improvement of the models' performance regarding
the East Asian Monsoon. While most CMIP3 models show a limited capacity in simulating the precipitation over East Asian
monsoon areas (Kai et al., 2009; Chen and Sun, 2013), the previous generation models of CMIP5 provided improvements
regarding observed spatial and temporal precipitation patterns (Seo et al., 2013). Nevertheless, CMIP5 models struggle to
reproduce rainfall bands around 30°N as well as the northward shift of the western North Pacific subtropical high (Huang
et al., 2013).
Further progress has been made by CMIP6 models that outperform their predecessors regarding the EAM in past periods
(Jiang et al., 2020; Xin et al., 2020; Yu et al., 2023). These improvements are related to the reduced biases in the sea surface





temperature (SST) over the Northwestern Pacific Ocean and better spatial resolution (Xin et al., 2020). In general, the CMIP6
models have reliable abilities in capturing the main characteristics of the East Asian monsoon, including the spatial distribution
of temperature and precipitation over China and the interannual variation (Xin et al., 2020; Masson-Delmotte et al., 2021).
However GCMs simulate 16-80% more national rainfall compared to observations during 1979-2005 (Jiang et al., 2020).
Previous studies have compared CMIP5 and CMIP6 models for past periods (Jiang et al., 2020; Xin et al., 2020; Yu et al.,
2023) or evaluated the changes of EASM in observations and CMIP6 models for 1979-2010 (Park et al., 2020) or analysed
the inter-model spread for 1979-2014 (Huang et al., 2022). Other studies have analysed the CMIP6 projections for the EAM
but only in the context of the global monsoon (Moon and Ha, 2020; Chen et al., 2020; Wang et al., 2020) and Asia monsoon
(Ha et al., 2020) neglecting e.g. regional model performance. To the best of the authors' knowledge, no study has put the
focus on the EAM providing detailed insight into projections for the EASM seasonal mean, its interannual variability as
well as the occurrence of extremely wet seasons for different time periods in the future under different emission scenarios.
Besides, we provide the central projections for China specifically, as highly relevant to policy makers. Here, we use the latest
generation of climate models in order to update the projected changes of the EAM rainfall under different socioeconomic
scenarios throughout the 21st century. For this purpose, we compare the available models and choose the ones with the best
performance for the further analysis. Section 2 provides a brief overview of the underlying climate model data and the Methods.
In Subsection 3.1, we divide the available models according to their performance in modeling the EASM in an historic period
in two groups. Subsection 3.2 presents the results of the mean summer monsoon precipitation, while Subsection 3.3 focuses on
the long-term trend of interannual variability and Subsection 3.4 provides further insights regarding the frequency of extremely
wet seasons. The results are discussed and concluded in Section 4.
## 2   Methods
In this study, we use 34 CMIP6 models that were available for the historic period (1850-2014) as well as for the future period
(2015-2100) under SSP5-8.5 in ScenarioMIP (O'Neill et al., 2016; Tebaldi et al., 2020). Per model center, a maximum of two
model configurations is used in order not to create bias. We use four scenarios (SSP1-2.6, SSP2-4.5, SSP3-7.0, SSP5-8.5) that
are based on different socioeconomic pathways with their associated greenhouse gas emissions as well as aerosol pollution
levels. These pathways are then translated into the resulting forcing levels (Van Vuuren et al., 2014; O'Neill et al., 2017). Table
1 provides an overview of the availability of the models in the different scenarios. The resolution of the native grids in which
the simulations were run are presented in Table A1 ranging from 2.5 to 500km. For the analysis, we regrid the model grids
to uniform 1°longitude x 1°latitude grids by first order conservative remapping. We use one ensemble member per model (if
available r1i1p1f1). Following existing literature, we focus on the land area in 20-50°N and 100-150°E to cover the East Asian
Monsoon (See Fig. A1). We obtain mean rainfall by averaging the monthly rainfall data during the summer monsoon season
from June to August. We use the future period from 2081-2100 and compare it to the reference period 1995-2014 in accordance
with the IPCC guidelines (Masson-Delmotte et al., 2021). For the analysis of the interannual variability and the occurrence of
extremely wet seasons, we compare 2050-2100 to 1965-2015 in order to have longer time periods and robuster results. For the



**Table 1.** Overview of data availability for the 34 models used in the study (precipitation/temperature). Only those models are selected for which data for historic period and SSP5-8.5 was available at the time of the study. Y indicates availability, N marks models that were not available for the scenario.

| Modeling Center (Group) | Model | SSP1-2.6 | SSP2-4.5 | SSP3-7.0 | SSP5-8.5 |
|---|---|---|---|---|---|
| Research Center for Environmental Changes, Academia Sinica (AS-RCEC) | Tai-ESM1 | N/Y | N/N | N/N | Y/Y |
| Alfred Wegener Institute (AWI) | AWI-CM-1-1-MR | Y/Y | Y/Y | N/N | Y/Y |
| Beijing Climate Center, China Meteorological Administration (BCC) | BCC-CSM2-MR | Y/Y | Y/Y | Y/Y | Y/Y |
| Chinese Academy of Meteorological Sciences (CAMS) | CAMS-CSM1-0 | Y/Y | Y/Y | Y/Y | Y/Y |
| LASG, Institute of Atmospheric Physics, Chinese Academy of Sciences (CAS) | FGOALS-f3-L | Y/Y | Y/Y | Y/Y | Y/Y |
| | FGOALS-g3 | Y/Y | Y/Y | Y/Y | Y/Y |
| Centre for Climate Change Research, Indian Institute of Tropical Meteorology (CCCR-IITM) | IITM-ESM | N/Y | N/N | N/N | Y/Y |
| Canadian Centre for Climate Modelling and Analysis (CCCma) | CanESM5 | Y/Y | Y/Y | Y/Y | Y/Y |
| | CanESM5-CanOE | Y/Y | Y/Y | N/N | Y/Y |
| Euro-Mediterranean Centre for Climate Change (CMCC) | CMCC-ESM2 | N/Y | N/N | N/Y | Y/Y |
| | CMCC-CM2-SR5 | N/Y | N/N | N/N | Y/Y |
| Centre National de Recherches Météorologiques/ Centre Européen de Recherche et Formation Avancées en Calcus Scientifique (CNRM-CERFACS) | CNRM-ESM2-1 | Y/Y | Y/Y | Y/N | Y/Y |
| | CNRM-CM6-1 | N/N | N/N | N/N | Y/Y |
| Commonwealth Scientific and Industrial Research Organisation (CSIRO) | ACCESS-ESM1-5 | Y/Y | Y/Y | Y/Y | Y/Y |
| Commonwealth Scientific and Industrial Research Organisation, ARC Centre of Excellence for Climate System Science (CSIRO-ARCCSS) | ACCESS-CM2 | Y/Y | Y/Y | Y/Y | Y/Y |
| EC-Earth-Consortium | EC-Earth3 | Y/Y | Y/Y | Y/Y | Y/Y |
| | EC-Earth3-CC | N/N | N/N | N/N | Y/Y |
| Energy Exascale Earth System Model Project (E3SM-Project) | E3SM-1-1 | N/N | N/N | N/N | Y/Y |
| First Institution of Oceanography (FIO-QLNM) | FIO-ESM-2-0 | Y/Y | Y/Y | N/N | Y/Y |
| Institute of Numerical Mathematics (INM) | INM-CM4-8 | Y/Y | Y/Y | Y/Y | Y/Y |
| | INM-CM5-0 | Y/Y | Y/Y | Y/Y | Y/Y |



| Modeling Center (Group) | Model | SSP1-2.6 | SSP2-4.5 | SSP3-7.0 | SSP5-8.5 |
|---|---|---|---|---|---|
| Institut Pierre Simon Laplace (IPSL) | IPSL-CM6A-LR | Y/Y | Y/Y | Y/Y | Y/Y |
| Japan Agency for Marine-Earth Science | MIROC6 | Y/Y | Y/Y | Y/Y | Y/Y |
| and Technology/ Atmosphere and Ocean | MIROC-ES2l | Y/Y | Y/Y | Y/Y | Y/Y |
| Research Institute, University of Tokyo | | | | | |
| (MIROC) | | | | | |
| Met Office Hadley Centre (MOHC) | UKESM1-0-LL | Y/Y | Y/Y | Y/Y | Y/Y |
| Max Planck Institute for Meteorology | MPI-ESM1-2-LR | Y/Y | Y/Y | Y/Y | Y/Y |
| (MPI-M) | | | | | |
| Meteorological Research Institute (MRI) | MRI-ESM2-0 | Y/Y | Y/Y | Y/Y | Y/Y |
| National Center for Atmospheric Re- | CESM2 | Y/Y | Y/Y | N/N | Y/Y |
| search (NCAR) | CESM2-WACCM | Y/Y | Y/Y | Y/Y | Y/Y |
| Norwegian Climate Center (NCC) | NorESM2-MM | Y/Y | Y/Y | Y/Y | Y/Y |
| National Institute of Meteorological | KACE-1-0-G | Y/Y | Y/Y | N/N | Y/Y |
| Sciences-Korea Met. Administration | | | | | |
| (NIMS-KMA) | | | | | |
| NOAA Geophysical Fluid Dynamics Lab- | GFDL-CM4 | N/N | Y/Y | N/N | Y/Y |
| oratory (NOAA-GFDL) | GFDL-ESM4 | Y/Y | Y/Y | Y/Y | Y/Y |
| Nanjing University of Information Sci- | NESM3 | Y/Y | Y/Y | N/N | Y/Y |
| ence and Technology (NUIST) | | | | | |
| | Number of models per scenario | 26/30 | 27/27 | 20/20 | 34/34 |

evaluation of the model, we use W5E5 reanalysis data (Lange, 2019) with a native grid of 0.5°longitude x 0.5°latitude grid
during the reference period and regrid it to the 1°longitude x 1°latitude grid. This data set is based on the WATCH Forcing Data
methodology applied to ERA5 data (WFDE5; Cucchi et al. (2020); Weedon et al. (2011)) and ERA5 reanalysis data (Hersbach
et al., 2019, 2020).

## 3 Results

### 3.1 Model comparison

To evaluate the models' capacity in capturing the seasonal rainfall of the EASM in the past, we compare the mean seasonal
rainfall to W5E5 reanalysis data in the period 1995-2014. The historical rainfall in the reanalysis data is $4.7 \pm 0.3$ mm/day.
While 16 out of 34 models are able to capture the historical mean within plus/minus two standard deviations, 14 models have a



tendency to overestimate and 4 models to underestimate the mean (See Fig. 1). The mean of the models range from 3.4 mm/day
(CAMS-CSM1-0) to 6.6 mm/day (INM-CM4-8). The models EC-Earth-3 and MPI-ESM1-2-LR capture the mean rainfall best.
The standard deviations of the model range from 0.2 mm/day (IITM-ESM) to 0.4 mm/day (INM-CM5-0). In this study, models
within two standard deviations are called group A models, the remaining ones group B models. The results in this study are
shown for group A models in the Results section, and in the Appendix for group B models.
The rainfall during the EAM is strongest along coastal regions, particularly in South and East China, the Korean peninsula,
as well as Japan and Taiwan (See Fig. 2). From the 16 group A models, most are able to reproduce major spatial patterns
including the rainfall in South China (Fig. 2). Regarding the Korean peninsula and Taiwan, the models have a tendency to
underestimate the local rainfall. Japan is captured reasonably well. The results for the individual models are shown in Fig. 3
and Fig. A1. Other studies focusing on the model evaluation provide further insides regarding the CMIP6 models' performance
for the EASM, e.g. Jiang et al. (2020).

## 3.2  Seasonal mean rainfall

In order to analyse the long-term trend of the EASM under climate change, we provide the time series between 1850-2100 for
all models in group A (Fig. 4) under four emission scenarios. Fig. B1 shows the time series averaged over all models including
group A and group B. The multi-model mean time series captures the decrease of rainfall in the second half of the 20th century
resulting from increasing aerosol pollution. The group A models show a stronger decrease in that time period compared to
group B models. This is followed by a rainfall increasing trend in the 21st century in all scenarios. The positive slopes in
the scenarios vary, potentially depending on the forcings resulting from the underlying socioeconomic pathway, particularly
aerosols and greenhouse gas emissions. High levels of development and the focus on health and environmental concerns in
SSP1 and SSP5 result in reduced air pollution emissions in the medium and long term. SSP2 has similar tendencies but
slower implementation and SSP3 is characterized by weak aerosol control and slow development of air pollution policies. This
explains that rainfall raises slower in SSP3 in the first half of the 21st century compared to other emission scenarios.
The time series for individual models under SSP5-8.5 are shown in Fig. 5 for group A and Fig. C1 for group B. Most group
A models reproduce the reducing the EASM monsoon rainfall in the second half of the 20th century. In the 21st century, all
group A models show a positive rainfall trend. Apart from MIROC-ES2L and CAMS-CSM1-0, also all group B models project
increasing rainfall during the 21st century.
To analyse the change in rainfall until the end of the 21st century, we calculate the difference in JJA rainfall from 2081-2100
compared to the reference period 1995-2014 for the four SSPs. Under SSP5-8.5, SSP3-7.0 and SSP2-4.5 all models project
an increase. Under SSP1-2.6, 25 out of 26 models project an increase in EASM rainfall. The increase differs between the
underlying emission scenarios: Under SSP5-8.5, the increase is 17.2% for the multi-mean of group A models (12.7% for group
B models). The largest increase in group A models is projected by KACE-1-0-G to be 28.7%, the smallest increase is projected
by AWI-CM-1-1-MR with 6.% (Fig. 6). Under SSP3-7.0, the group A models project an average increase of 12.7% (8.0% for
group B; Fig. 7). Under SSP2-4.5, the increase projected by group A models is 11.9% (8.6% for group B; Fig. 8) and under
SSP1-2.6, it is 11.2% (6.6%; Fig. 9). The group A models project stronger increases in the EASM rainfall. Besides, it has to



**Figure 1.** Mean Rainfall of the East Asian Summer Monsoon from June-September (mm day$^{-1}$) over the region displayed in Fig. A1 from 34 CMIP6 models. The vertical line mark the mean monsoon rainfall from W5E5 renalysis data (continuous line) plus/minus two standard deviations (dashed line). Circles with error bars represent mean plus/minus one standard deviation for each individual climate model during the same period.



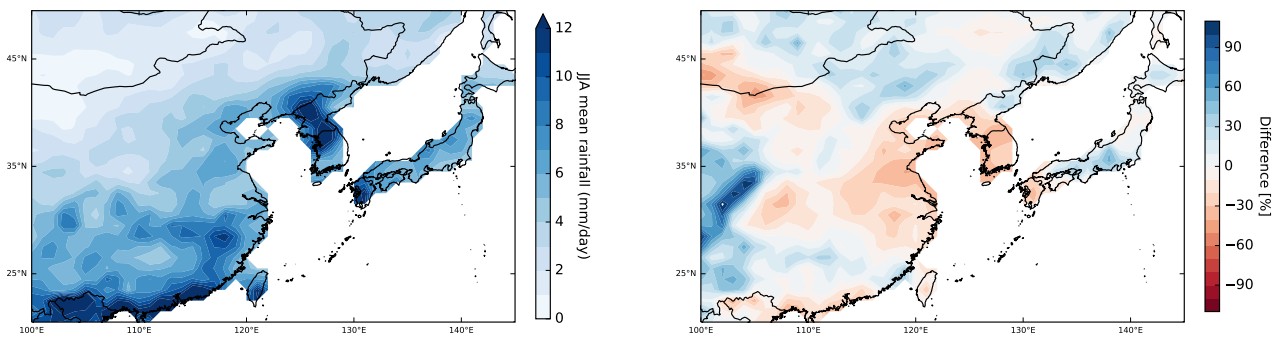

**Figure 2.** Left: Spatial distribution of EAM averaged over the period 1995-2014 from W5E5 reanalysis data. Right: Difference [%] between reanalysis data and multi-model-mean of the 16 group A models for the same time period.

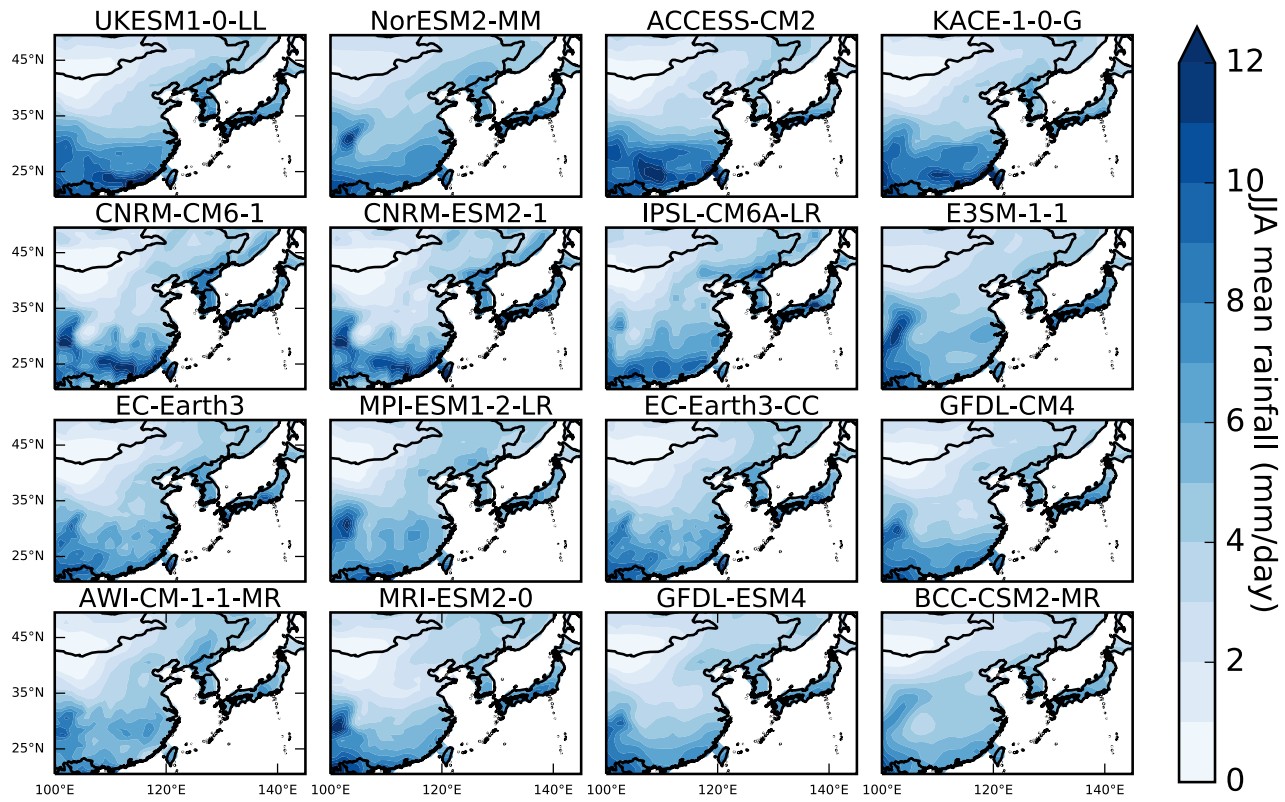

**Figure 3.** Spatial distribution of EASM averaged over the period 1995-2014 from the 16 models in group A.



be noted that the projections of all scenarios lie within the uncertainty ranges of the other scenarios. Further details regarding
other periods (2021-2040, 2041-2060, 2061-2080) can be found in Table 2. Regarding the monsoon change only over China,
the increase projected by group A models is even stronger: Under SSP1-2.6 the monsoon rainfall intensifies by 12.1%, under
SSP2-4.5 by 12.7%, under SSP3-7.0 by 14.1% and under SSP5-8.5 by 19.1% in multi-model average.
The spatial change in EASM rainfall between 2081-2100 and 1995-2014 based on the group A multi-model mean is shown
in Fig. 10 for SSP5-8.5. The rainfall in the entire EASM region is projected to increase in multi-model mean, particularly in
Taiwan, South-East China as well as North Korea and adjacent regions. The majority of group A models coincide in the larger
scale rainfall change pattern. Though, some models project regional decrease of rainfall in different areas (Fig. D1). The results
for group B models are shown in Fig. E1. Fig. F1 shows the multi-model mean of spatial changes for the EASM under the four
different scenarios only for the models that are available for all four scenarios. The regions with intensifying rainfall coincide
with the areas under SSP5-8.5, though the intensity varies according to the underlying forcing.
Besides, we analyse the dependence of EASM rainfall on global mean temperature (GMT). The multi-model mean indicates
a linear relationship relatively independent of the underlying emission scenario (Fig. 11). The projected average increase
in daily rainfall during the monsoon season is 0.14mm per degree of global warming. This refers to an increase in EASM
rainfall of 3.0% (1.4-4.6%) per degree GMT increase. The increase ranges from 0.06mm/day to 0.22mm/day depending on the
providing model.

**Table 2.** Multi-model mean changes [%] in JJA rainfall from different future periods compared to reference period 1995-2014 (group A/ group B/ all).

| Scenario | 2021-2040 | 2041-2060 | 2061-2080 | 2081-2100 |
|----------|-----------|-----------|-----------|-----------|
| SSP1-2.6 | 4.7/2.9/3.7 | 8.3/5.5/6.8 | 10.2/6.0/7.9 | 11.2/6.6/8.8 |
| SSP2-4.5 | 4.9/2.0/3.4 | 7.8/4.7/6.2 | 9.1/6.2/7.6 | 11.9/8.6/10.2 |
| SSP3-7.0 | 3.4/0.3/1.9 | 5.4/2.4/3.9 | 9.4/4.0/6.7 | 12.7/8.0/10.4 |
| SSP5-8.5 | 5.9/2.6/4.2 | 9.2/5.3/7.1 | 13.0/8.6/10.7 | 17.2/12.7/14.8 |

## 3.3   Interannual variability
Furthermore, we analyse the interannual variability of the EASM rainfall. For this purpose, we remove the nonlinear trend
obtained by the singular spectrum analysis (see Fig. 5 and C1). We use the percentage changes in standard deviation between
2050-2100 and 1965-2015. Under SSP5-8.5, 15 of the 16 group A models project an increase of interannual variability with a
multi-mean of 31.4% ranging from -1.5% to 31.4% (Fig. 12). Under SSP3-7.0, 7/10 group A models project an increase with
an average of 10.6% (-20.0% to 34.3%; Fig. G1). Under SSP2-4.5, 10/13 project increasing variability of 10.4% (-14.0% to
30.8%;Fig. H1) and under SSP1-2.6, an increase is projected by 7/12 group A models with a multi-model average of 7.0%
(-24.8% to 27.5%; Fig. I1). With stronger emission scenarios, the increase of interannual variability is stronger with more
models coinciding in the sign of the change.





**Figure 4.** Time series of EASM (mm $d^{-1}$) for the period 1850-2100 based on the multi-model mean of the 16 models in group A relative to the period 1995-2014. The time series is based on the 20-years-running-mean of the individual models. The shading marks the range of plus/minus one standard deviation. Availability of the models in accordance with Table 1.

### 3.4 Extremely wet seasons

We use the 90th percentile for the period 1965-2015 in order to define extremely wet monsoon seasons. Thus, per definition 5 out of 50 years were extremely wet during the 50-years period from 1965-2015. Under SSP5-8.5, the number of extremely wet monsoon seasons will increase by a factor of 6.5 until 2050-2100. Respectively, 32.4 years are expected to be extremely wet in 2050-2100 with individual group A model projections ranging from 14 to 46 out of 50 seasons. Under SSP3-7.0, the multi-model mean projection is 26.1 ranging from 7 to 41 extremely wet seasons. Under SSP2-4.5, 25.5 seasons in the future period are projected to be extremely wet ranging from 7 to 36 and under SSP1-2.6 the multi-model mean projection is 25.6 ranging from 8 to 37 seasons. The increase over time is shown in Fig. 13.

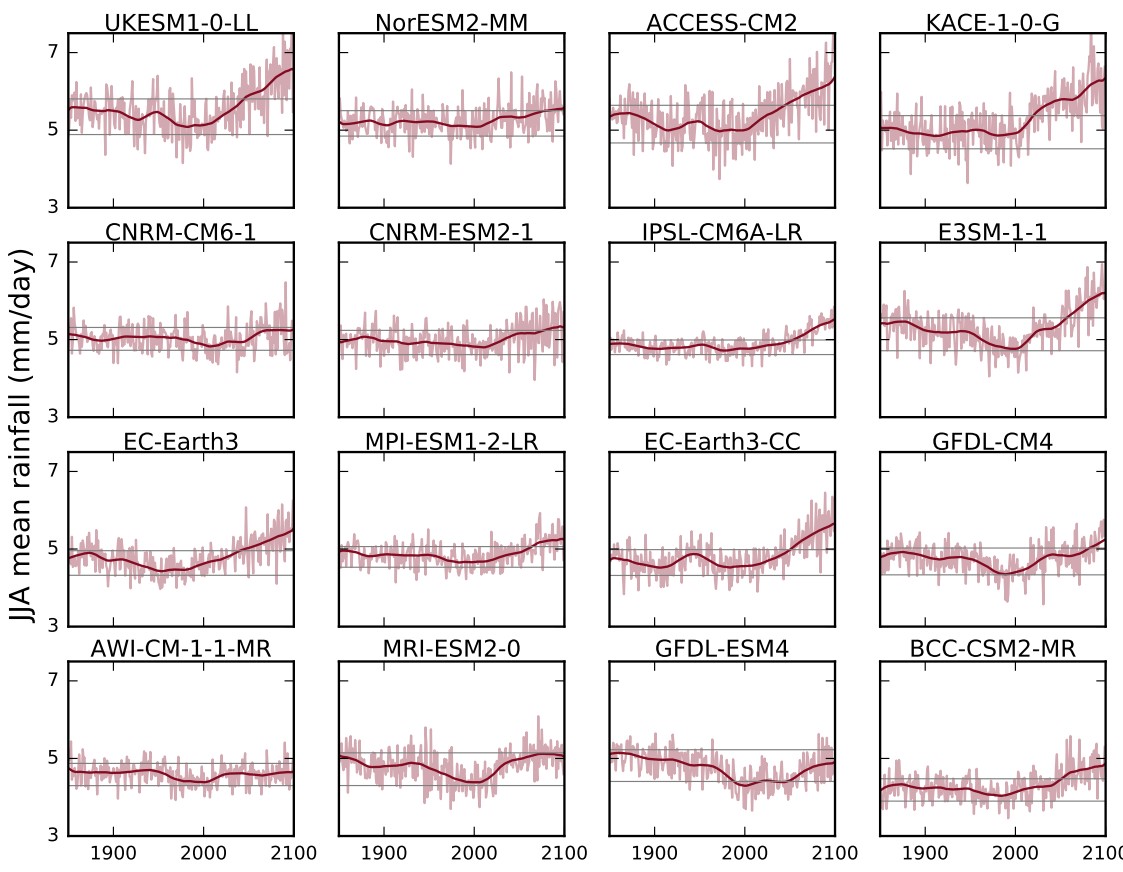

**Figure 5.** Time series of EASM (mm/day) for the period 1850-2100 from the 16 models in group A. Light red lines represent the annual values; red lines mark the trend obtained from a singular spectrum analysis with a window size of 20 years. For the method, see Golyandina and Zhigljavsky (2013). The horizontal grey lines represent mean ± standard deviation for each model for the period 1850-2015.



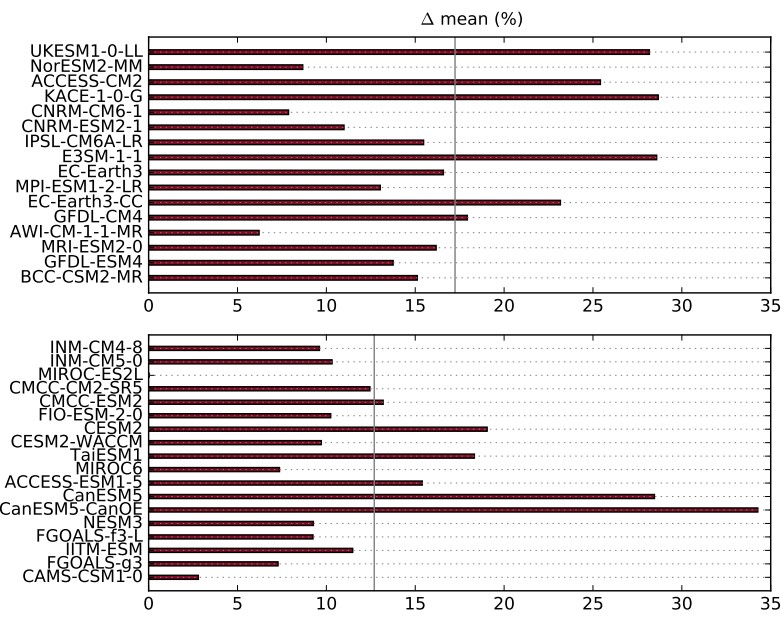

**Figure 6.** Changes [%] in JJA rainfall between 2081-2100 and 1995-2014 under SSP5-8.5. Upper panel shows group A models, the lower panel group B models. The vertical line marks the multi-model-mean for both groups.

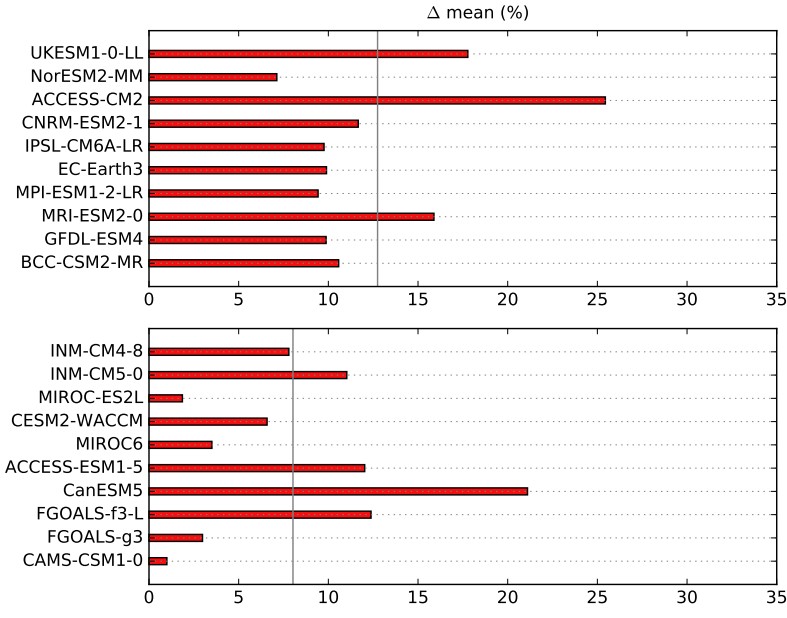

**Figure 7.** Changes [%] in JJA rainfall between 2081-2100 and 1995-2014 under SSP3-7.0. Upper panel shows group A models, the lower panel group B models. The vertical line marks the multi-model-mean for both groups.





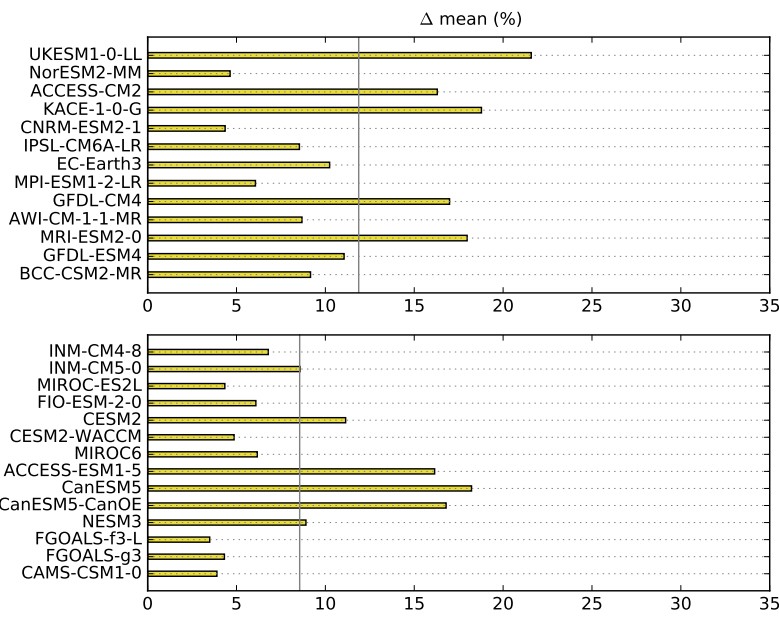

**Figure 8.** Changes [%] in JJA rainfall between 2081-2100 and 1995-2014 under SSP2-4.5. Upper panel shows group A models, the lower panel group B models. The vertical line marks the multi-model-mean for both groups.

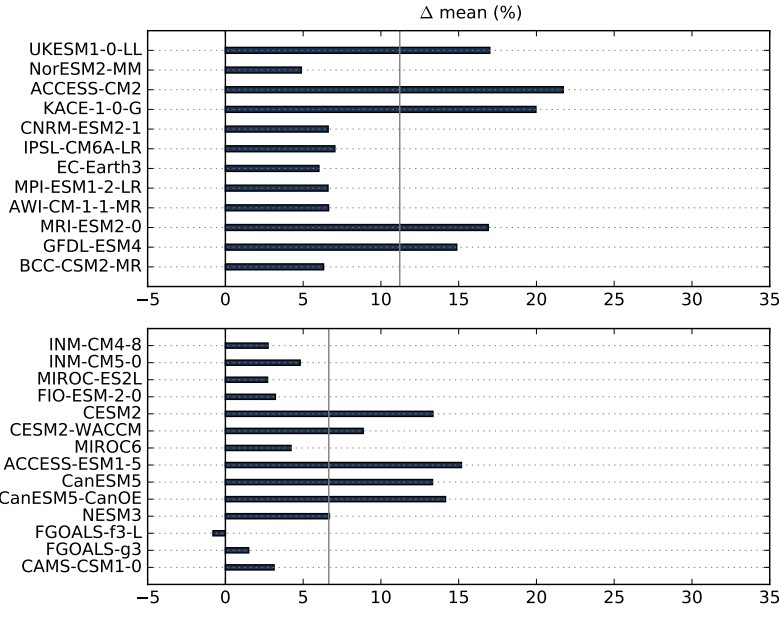

**Figure 9.** Changes [%] in JJA rainfall between 2081-2100 and 1995-2014 under SSP1-2.6. Upper panel shows group A models, the lower panel group B models. The vertical line marks the multi-model-mean for both groups.

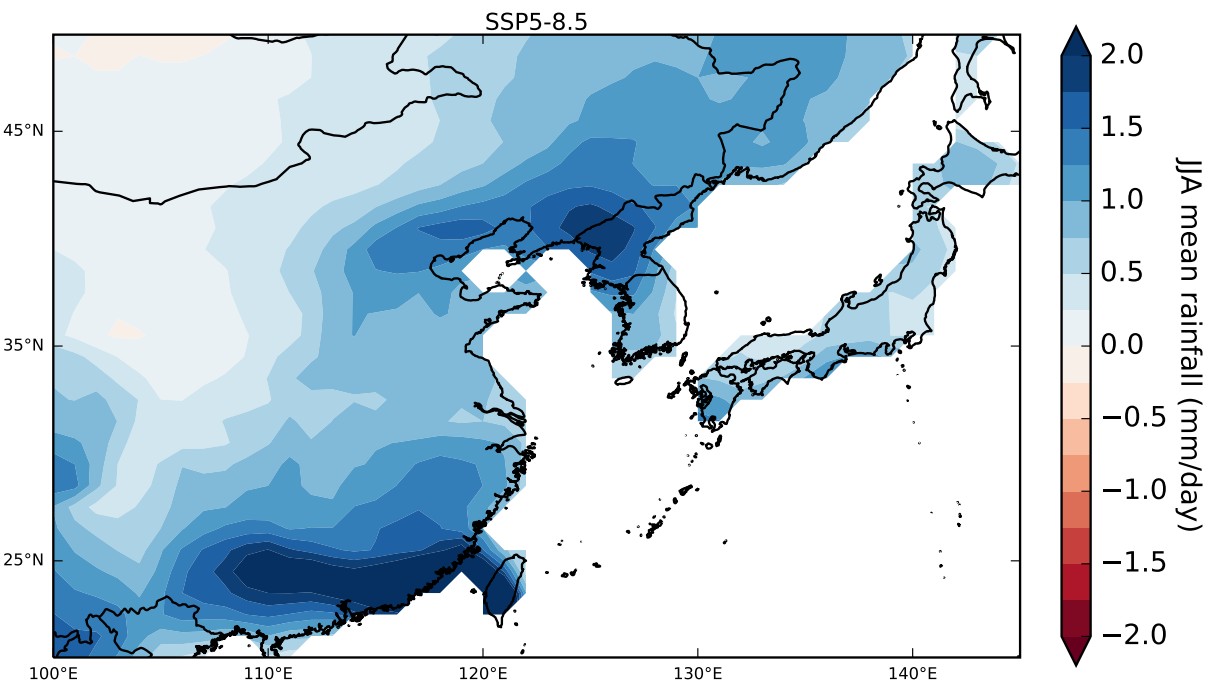

**Figure 10.** Spatial changes in JJA rainfall between 2081-2100 and 1995-2014 under SSP5-8.5 for multi-modeal mean of group A models. The individual model results are shown in Fig. D1 and Fig. E1.





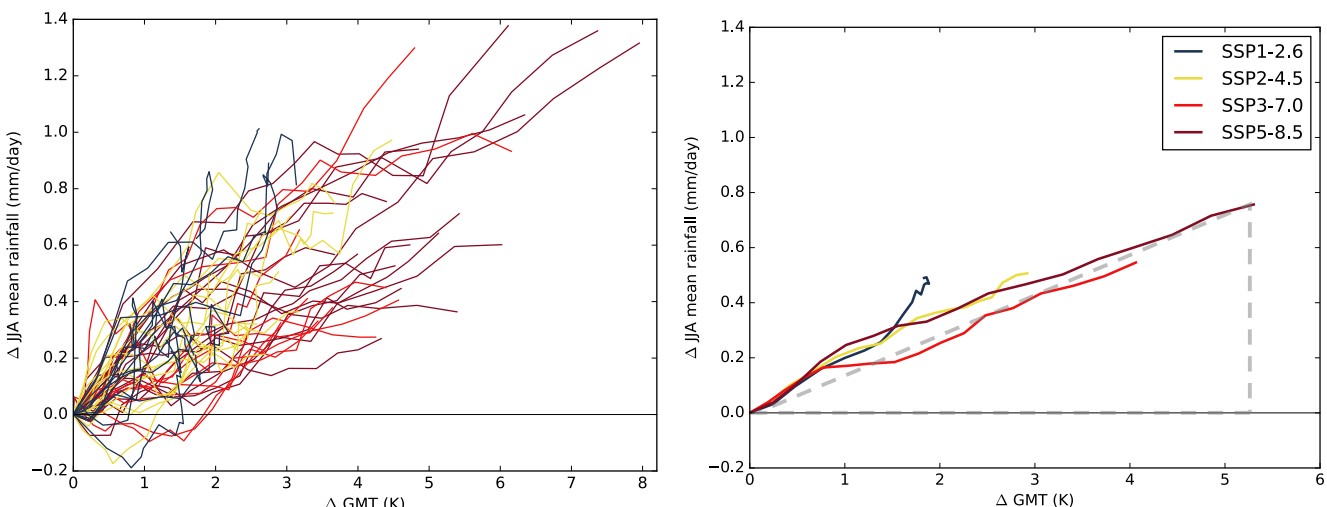

**Figure 11.** Change of EASM rainfall (mm/day) depending on change in global mean temperature (K) during the 21st century for all group A models (left) and their multi-model average (right). The change is shown based on 20-year periods (1995-2015, 2000-2020, 2005-2025,...). Dashed gray lines indicate the slope. The reference period is 1995-2014.

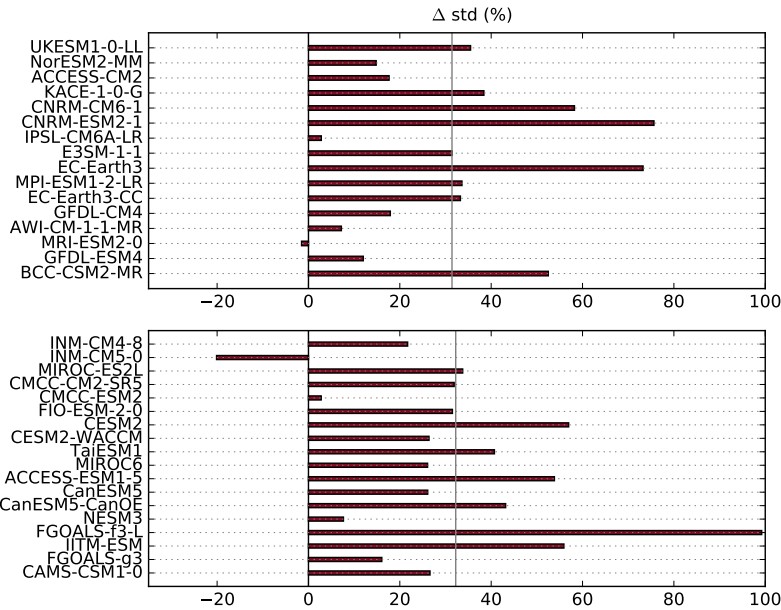

**Figure 12.** Change [%] of interannual variability between 2050-2100 and 1965-2015 for the EASM seasonal rainfall under SSP5-8.5. The upper panels show the group A models, the lower panels the group B models. The vertical line indicates the multi-model mean of the respective group.



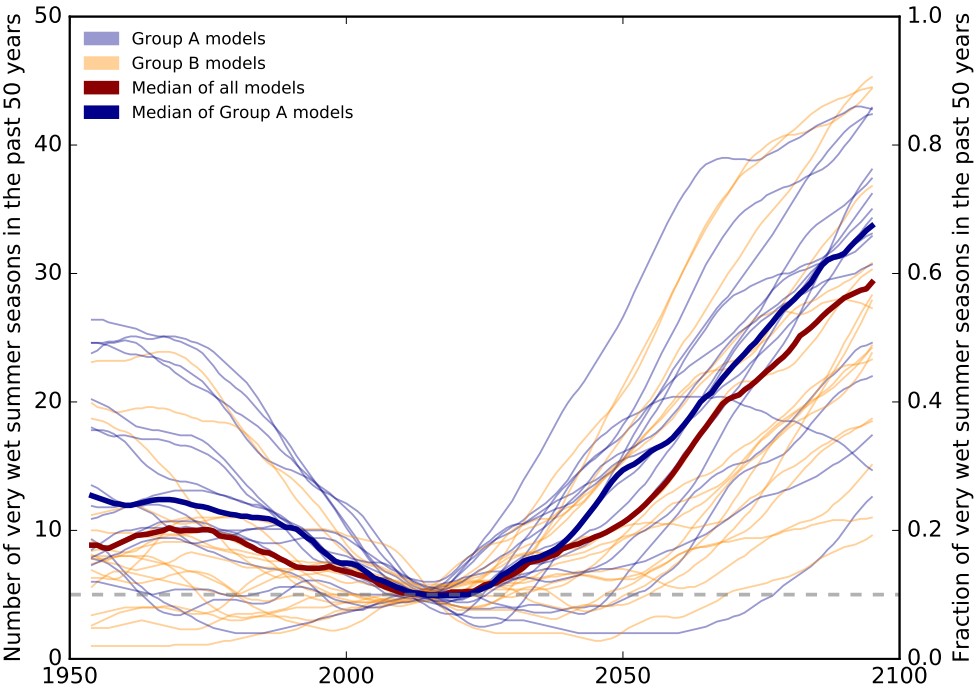

**Figure 13.** Increase of extremely wet monsoon seasons under unabated climate change (SSP5-8.5). Group A models are shown in blue, group B models in orange. The reference period is 1965-2015 where per definition 5 out of 50 years were extremely wet.

## 4 Discussion and Conclusion

In this study, we use 34 CMIP6 models in order to analyse their future projections under climate change regarding the East Asian Summer Monsoon. We identify models that capture the rainfall in 1995-2014 within two standard deviations as group A models and use them for our main analysis. The CMIP6 models have a tendency to overestimate the EASM rainfall which is in line with previous studies (Jiang et al., 2020). This is different from other Asian monsoon regions, e.g. in the Indian monsoon region models tend to underestimate the seasonal rainfall (Katzenberger et al., 2021, 2022). All group A models robustly project an increase of rainfall under all four emission scenarios. The projected multi-model mean increase until 2081-2100 is 17.2% under SSP5-8.5, 12.7% under SSP3-7.0, 11.9% under SSP2-4.5 and 11.2% under SSP1-2.6. The rainfall-intensifying tendency is also confirmed by the IPCC, AR6 classifying the increasing trend as 'highly certain' (Masson-Delmotte et al., 2021). The projected increase is also in line with CMIP5 projections, though even stronger increases are projected in CMIP6 (Qu et al., 2014; Chen and Sun, 2013; Kitoh et al., 2013). But it has to be noted, that there are differences in the methods between the studies, preventing direct comparison of the results. The projections for the near-term depend on the implementation and



efficiency of future air pollution control that is difficult to predict (Wilcox et al., 2020) adding uncertainty mainly for the period
2021-2040. The increase in rainfall will particularly contribute to rainfall in South East China, Taiwan as well as North Korea -
regions that are already experiencing a relatively strong monsoon. Thus the wet-regions-get-wetter dynamics is predominantly
confirmed for the EASM in line with CMIP5 results (Seo et al., 2013). Over China, the monsoon is projected to increase
by 12.1% under SSP1-2.6, under SSP2-4.5 by 12.7%, under SSP3-7.0 by 14.1% and under SSP5-8.5 by 19.1%. Per degree
of global warming, the monsoon is projected to increase by 0.14mm/day which refers to 3% of the rainfall in the reference
period. The intensification of the EASM is resulting from the combined effects of an enhanced evaporation due to increased
sea surface temperatures, increased water vapour as well as moist flux convergence induced by the (north) westward shift of
the North Pacific subtropical high (Seo et al., 2013; Qu et al., 2014). Additionally, the strengthening of the land-sea thermal
contrast under global warming contributes to the rainfall increase of Asian monsoon systems (Endo et al., 2018).
Besides, we analysed the interannual variability that is particularly important for societal and economic adaptation strategies,
defining the necessary interannual flexibility for agricultural irrigation, flooding management, etc. The interannual variability
is projected to increase by 7.0% under SSP1-2.6, 10.4% under SSP2-4.5, 10.5% under SSP3-7.0 and 31.4% under SSP5-8.5
from 1965-2015 to 2050-2100. Comparing the CMIP6 multi-mean results under SSP5-8.5 of 31.4% to CMIP3 results under
the respective A2 scenario, the projected increase in CMIP3 of 19% is considerably weaker (Lu and Fu, 2010). Additionally,
extremely wet monsoon seasons are projected to occur 6.5 times more often under SSP5-8.5 compared to the reference period.
The increase of interannual variability of the seasonal rainfall is accompanied by increasing interannual variability of the
western North Pacific subtropical high and East Asian upper-tropospheric jet (Lu and Fu, 2010). The projected changes in
the characteristics of the EASM are of high socioeconomic relevance and should be taken into account in the management
decisions for the 21st century.





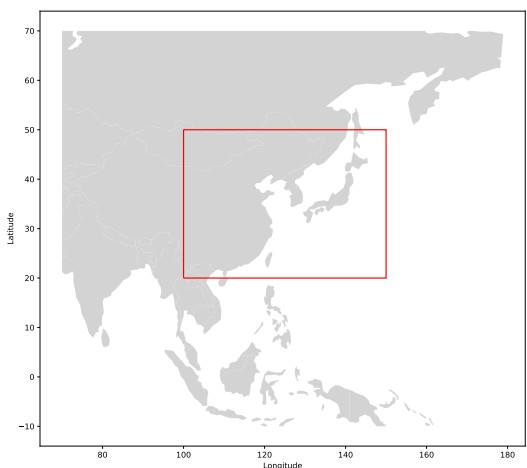

**Figure A1.** Area of the East Asian summer monsoon from 20-50°N and 100-150°E as covered in this study.



**Table A1.** Overview of the model resolutions of the native model grids in which the 34 CMIP models were run. For the analysis in this study, the models have been remapped to a 1°horizontal grid.

| Model | Atmosphere [km] | Land [km] | Ocean [km] |
|---|---|---|---|
| Tai-ESM1 | 100 | 100 | 100 |
| AWI-CM-1-1-MR | 100 | 100 | 25 |
| BCC-CSM2-MR | 100 | 100 | 50 |
| CAMS-CSM1-0 | 100 | 100 | 100 |
| FGOALS-f3-L | 100 | 100 | 100 |
| FGOALS-g3 | 250 | 250 | 100 |
| IITM-ESM | 250 | 250 | 100 |
| CanESM5 | 500 | 500 | 100 |
| CanESM5-CanOE | 500 | 500 | 100 |
| CMCC-ESM2 | 100 | 100 | 100 |
| CMCC-CM2-SR5 | 100 | 100 | 100 |
| CNRM-ESM2-1 | 250 | 250 | 100 |
| CNRM-CM6-1 | 250 | 250 | 100 |
| ACCESS-ESM1-5 | 250 | 250 | 100 |
| ACCESS-CM2 | 250 | 250 | 100 |
| EC-Earth3 | 100 | 100 | 100 |
| EC-Earth3-CC | 100 | 100 | 100 |
| E3SM-1-1 | 100 | 100 | 50 |
| FIO-ESM-2-0 | 100 | 100 | 100 |
| INM-CM4-8 | 100 | 100 | 100 |
| INM-CM5-0 | 100 | 100 | 50 |
| IPSL-CM6A-LR | 250 | 250 | 100 |
| MIROC6 | 250 | 250 | 100 |
| MIROC-ES2l | 500 | 500 | 100 |
| UKESM1-0-LL | 250 | 250 | 100 |
| MPI-ESM1-2-LR | 250 | 250 | 250 |
| MRI-ESM2-0 | 100 | 100 | 100 |
| GISS-E2-1-G | 250 | 250 | 100 |
| CESM2 | 100 | 100 | 100 |
| CESM2-WACCM | 100 | 100 | 100 |





| Model | Atmosphere [km] | Land [km] | Ocean [km] |
|---|---|---|---|
| NorESM2-MM | 100 | 100 | 100 |
| KACE-1-0-G | 250 | 250 | 100 |
| GFDL-CM4 | 100 | 100 | 25 |
| GFDL-ESM4 | 100 | 100 | 50 |
| NESM3 | 250 | 2.5 | 100 |

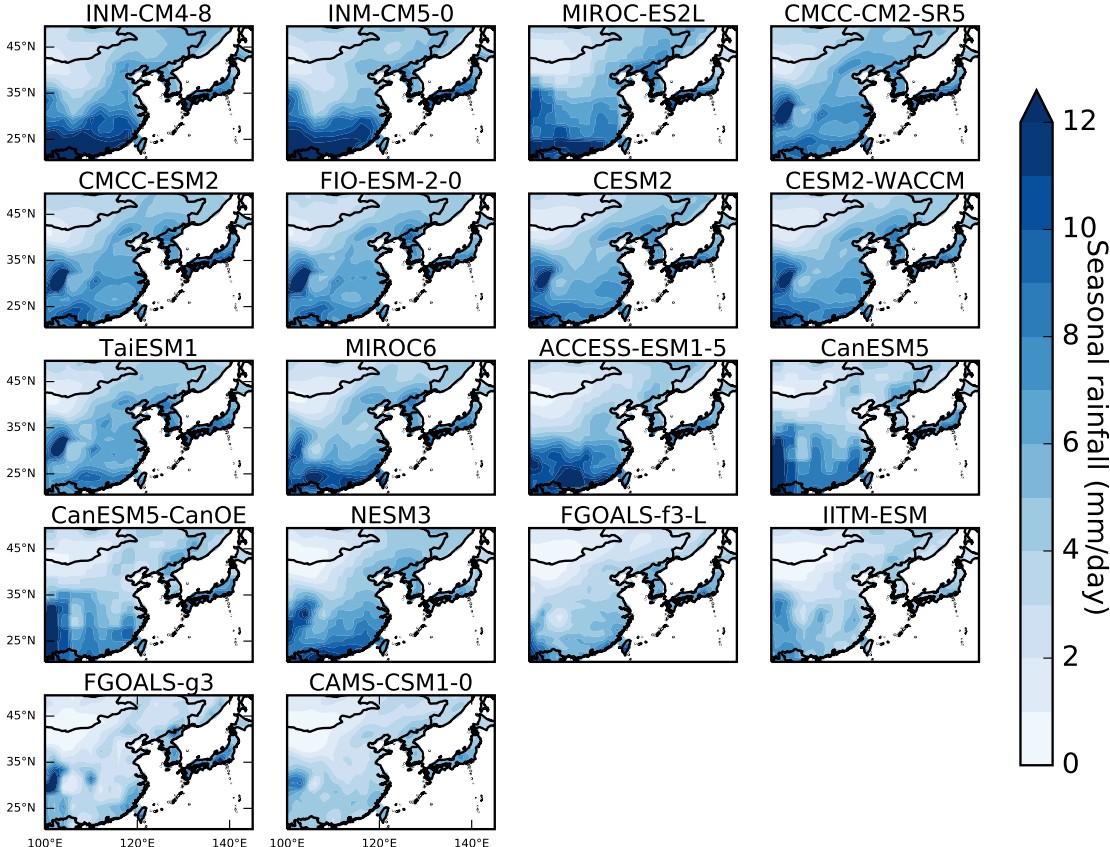

**Figure A1.** Spatial distribution of EAM averaged over the period 1995-2014 from the 18 models in group B.



**Figure B1.** As in Fig. 4 but including all 34 models from group A and B.



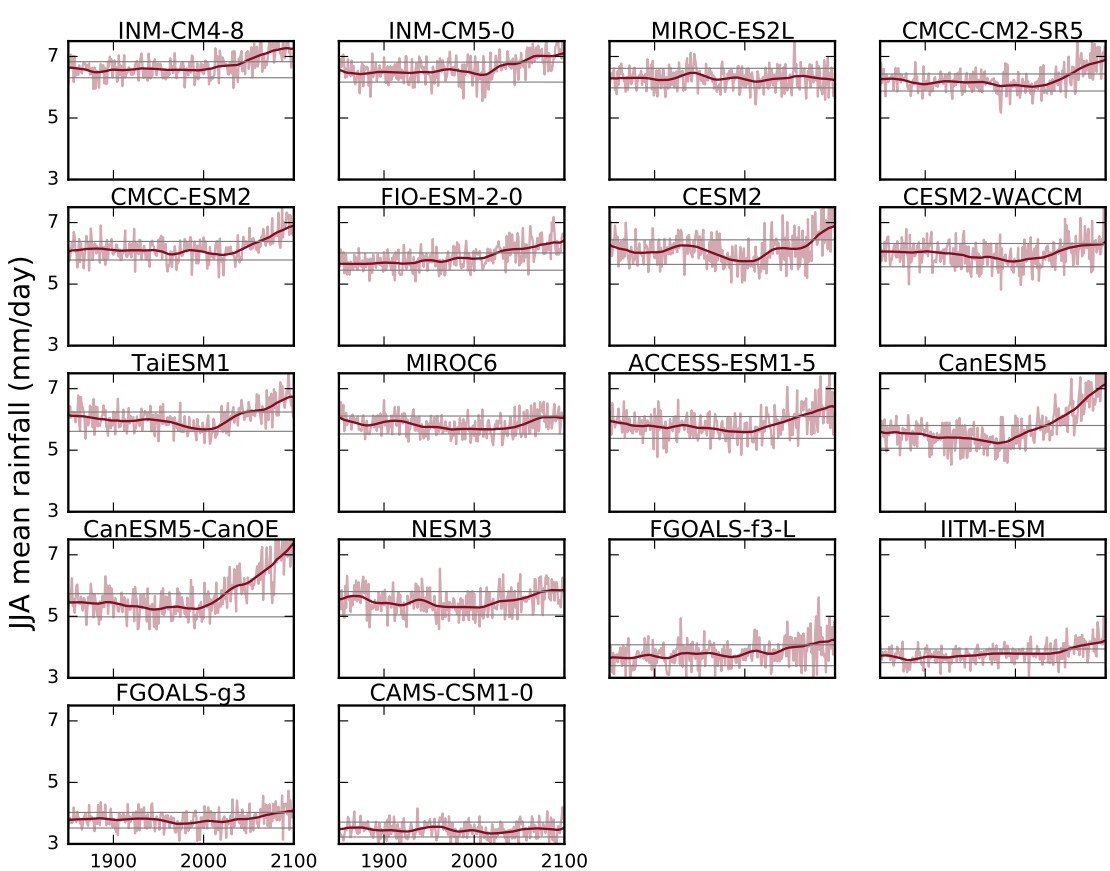

**Figure C1.** As in Fig. 5 but for the models in Group B.



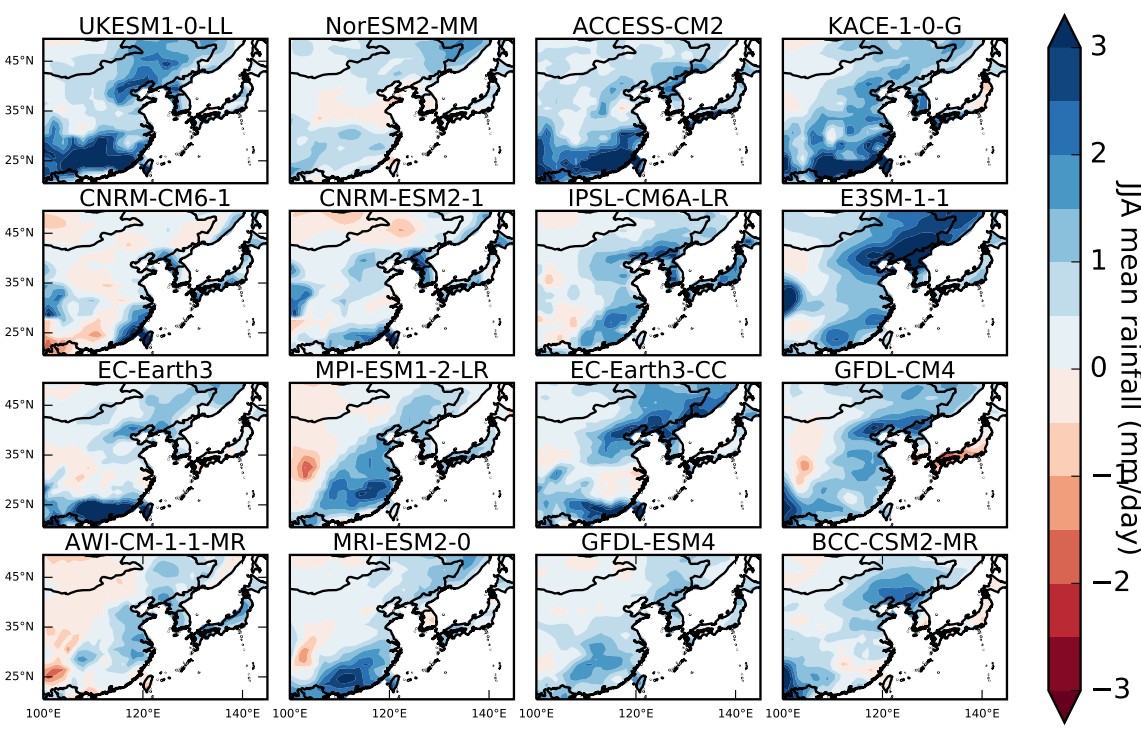

**Figure D1.** Spatial changes in JJA rainfall between 2081-2100 and 1995-2014 under SSP5-8.5 for group A models. The multi-model mean is shown in Fig. 10.

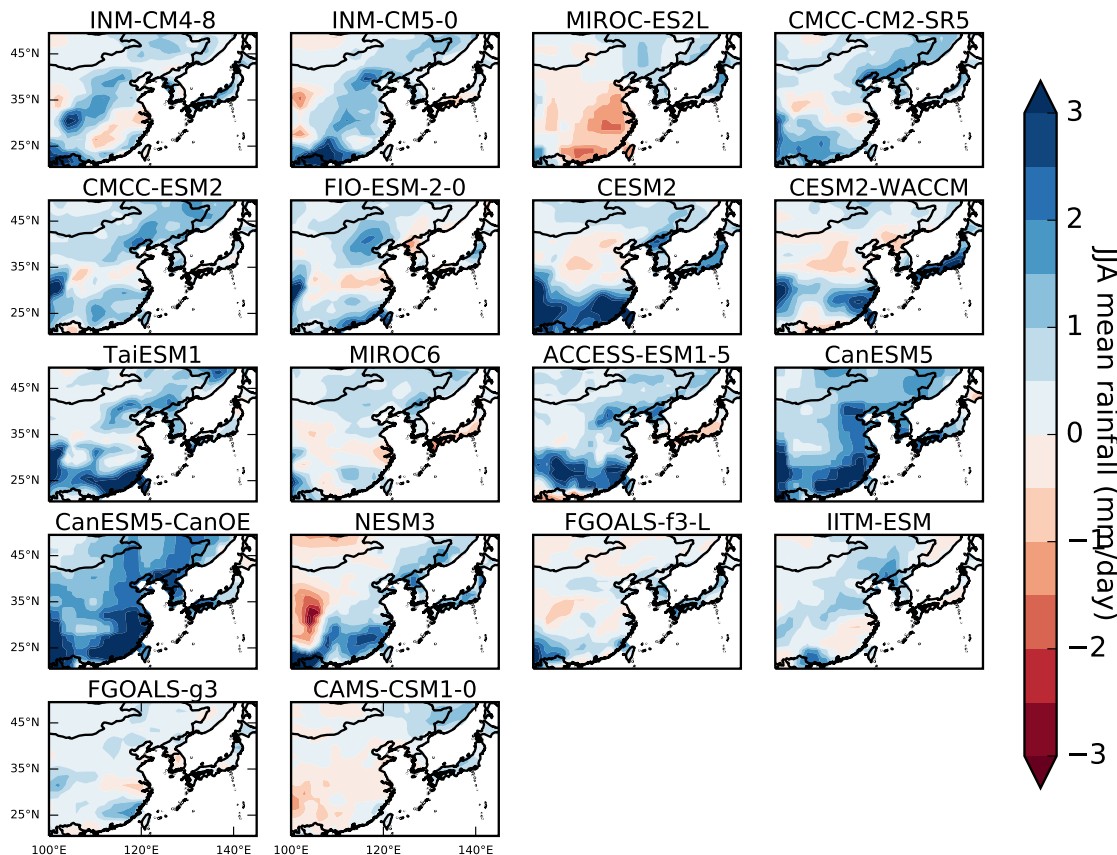

**Figure E1.** Spatial changes in JJA rainfall between 2081-2100 and 1995-2014 under SSP5-8.5 for group B models.

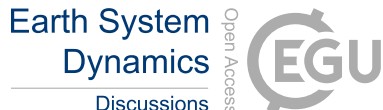

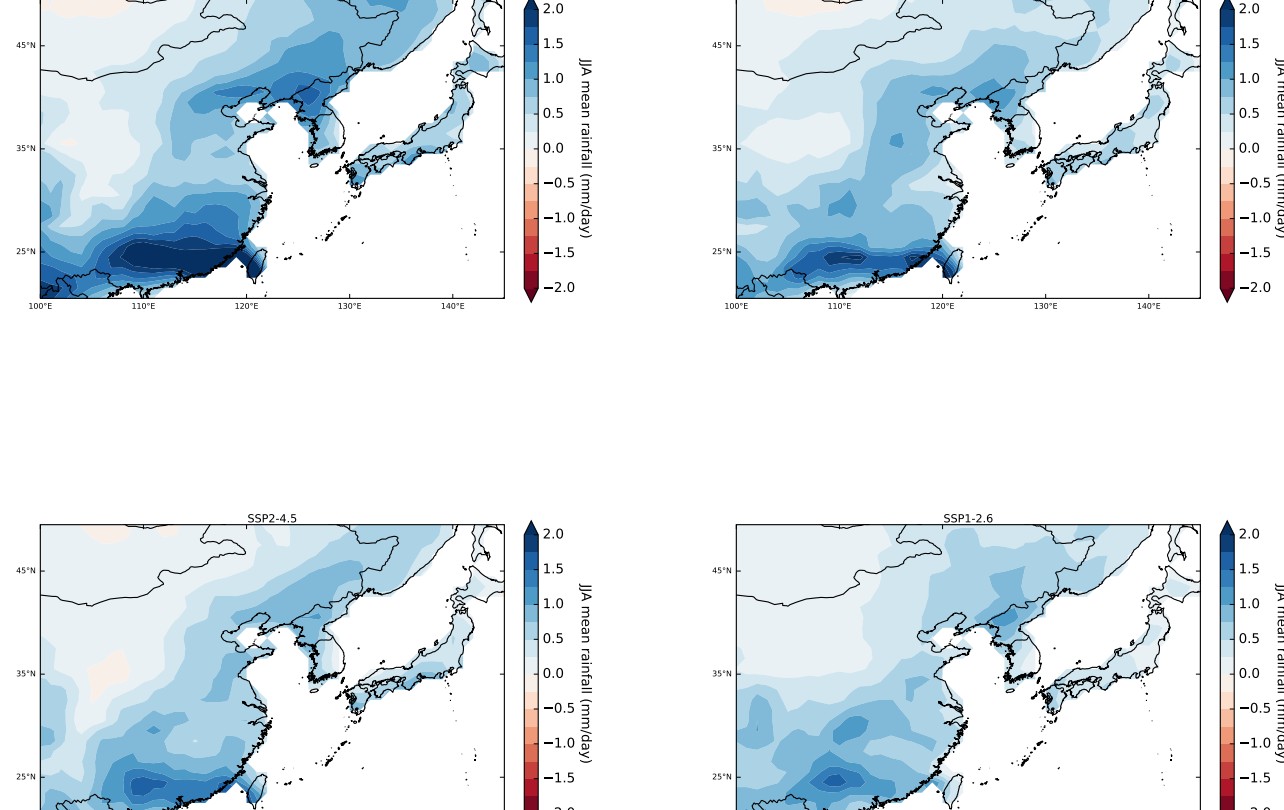

**Figure F1.** As in Fig. 10, but using only the group A models that are available for all four scenarios.

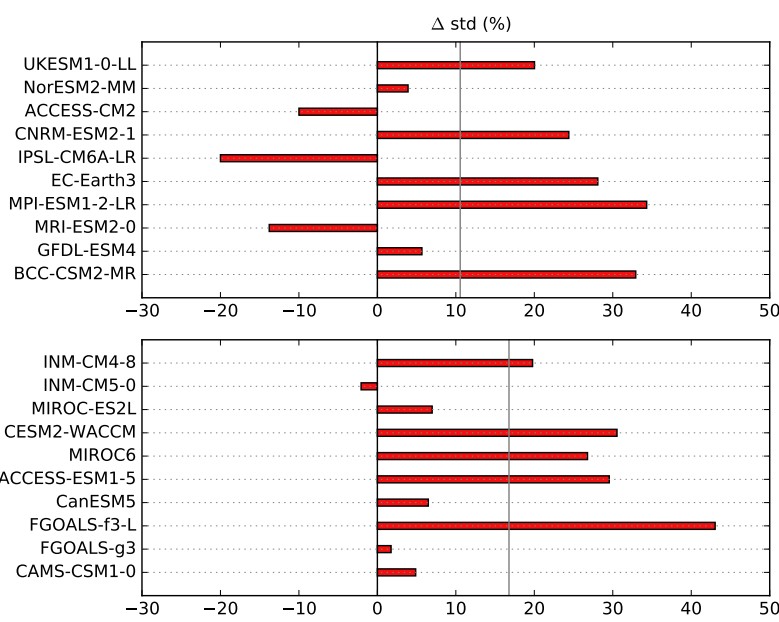

**Figure G1.** Change [%] of interannual variability between 2050-2100 and 1900-1950 for the EASM seasonal rainfall under SSP3-7.0. The upper panels show the group A models, the lower panels the group B models. The vertical line indicates the multi-model mean of the respective group.



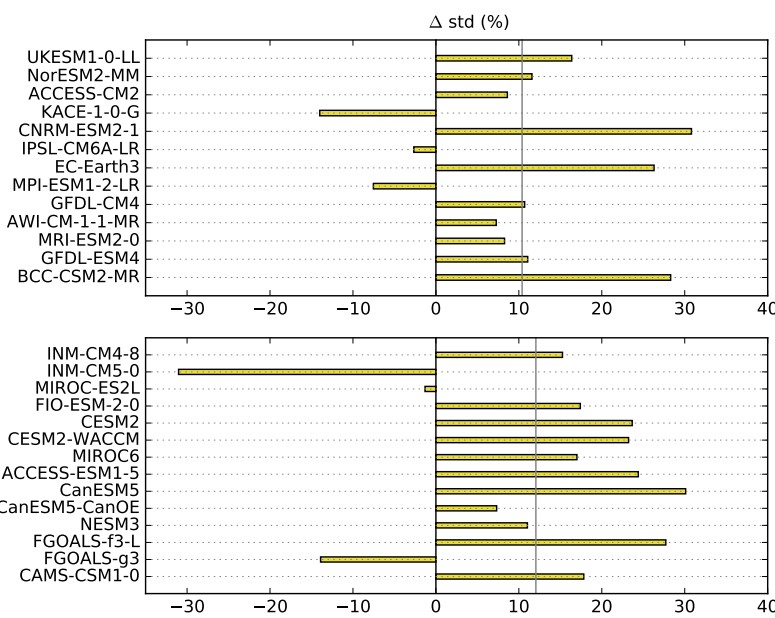

**Figure H1.** Change [%] of interannual variability between 2050-2100 and 1900-1950 for the EASM seasonal rainfall under SSP2-4.5. The upper panels show the group A models, the lower panels the group B models. The vertical line indicates the multi-model mean of the respective group.





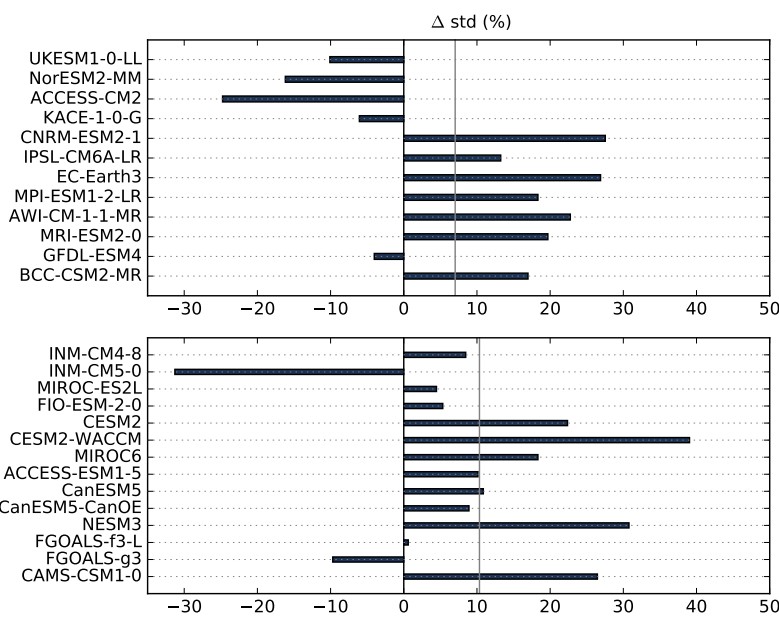

**Figure I1.** Change [%] of interannual variability between 2050-2100 and 1900-1950 for the EASM seasonal rainfall under SSP1-2.6. The upper panels show the group A models, the lower panels the group B models. The vertical line indicates the multi-model mean of the respective group.



**Statements and Declarations**
*Code and data availability.*  The data sets from CMIP6 simulations are available via the CMIP6 Search Interface: https://esgf-node.llnl.gov/
search/cmip6/ (last access: 31 March 2023) (WCRP). The relevant CMIP6 data extract as well as the underlying code is available in a private
github repository that will be made public and linked to zenodo when this article will be published.
*Funding:* The research was financially supported by the Heinrich-Boell Foundation who did not have any influence on the
study design, the data analysis or the interpretation of the results (nor any other influence).
*Author contributions.*  AL proposed the idea of this study. AK performed the analysis and wrote the paper. AK and AL discussed the results
and approved the final version.
*Competing interests.*  At least one of the (co-)authors is a member of the editorial board of Earth System Dynamics. The peer-review process
was guided by an independent editor, and the authors have also no other competing interests to declare.
*Acknowledgements.*  We acknowledge the World Climate Research Programme's Working Group on Coupled Modelling, which is respon-
sible for CMIP, and we thank the climate modelling groups for producing and making available their model output. Besides, we thank the
Copernicus Climate Change Service for providing the WFDE5 reanalysis data set.





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
