# Peer review of "Consistent increase of East Asian Summer Monsoon rainfall and its variability under climate change over China in 34 coupled climate models"

_Earth System Dynamics, 2023_

## Author Comment (AC1)

We thank the two anonymous reviewers for the valuable comments that contributed to a substantially improved version of the manuscript. The major changes resulting from the two reviewer reports obtained are:

- adding further model evaluation criteria (regarding interannual variability, spatial distribution, 850hPa wind) resulting in some changes in the selected model set

- focusing the results on the best performing models only, called TOP6 throughout the manuscript and adapting previous results accordingly

- focusing on area that additionally fulfills monsoon definition (JJA minus DJF rainfall exceeds 2 mm/day) and providing results only based on this area

- using observational data (GPCC) instead of reanalysis data (W5E5) for reference

- discussing underlying physical mechanisms for changes in mean circulation (by analysing wind 850 hPa)

- adding a significance measure to give insight into the robustness of the projections following IPCC standards

- adding a new subchapter regarding the wet bulb temperature projections

We are looking forward to further feedback from the reviewers.

**-----------------------**

Reviewer #1

**-----------------------**

Review of "Consistent increase of East Asian Summer Monsoon rainfall and its variability under climate change over China in 34 coupled climate models" by Anja Katzenberger and Anders Levermann

General Comments: In this paper, the authors examined the future changes in the mean precipitation over the EASM region as well as its variability under different emission scenarios. Their analysis suggests that both mean precipitation and its variability are increasing under all emission scenarios, with a stronger response under stronger emission scenarios. There were many studies using CMIP5/6 model simulations that looked at the precipitation changes over different monsoon regions of the globe. Also, they find that the "wet-gets-wetter" arguments holds true for EASM region. Overall, the types of analyses presented in this paper are useful for regional climate change assessments. I have some specific concerns that need to be addressed before accepting the article.

Specific comments:

1. My major concern in this paper is the way in which the models are grouped. I don't have any issues with the "Group A" models which have a mean precipitation withing +/- 2 std of the observations (reanalysis). However, the Group B consists of the models in both sides of the extremes. This means that when you take the ensemble

mean, you are averaging the outliers on two sides and as a result there may not be any use of grouping the models in this way. You may either focus on the Group A models or have three groups (one group each for outliers on each side).

In the revised manuscript, we added further selection criteria (STD, CRSME, WIND 850 hPa) resulting in an adapted group of selected models, called TOP6. We follow the reviewer's proposition to focus on the best performing models.

However, depending on the research question, it can also have advantages to generally use a larger ensemble member size including over- and underestimating models. Particularly because the overestimating models compensate for the underestimating models resulting in reasonable multi-model performance. See e.g. Sing & AchutaRao (2018): Quantifying uncertainty in twenty-first century climate change over India.

2. It would be interesting to see the seasonal mean circulation changes as well. This will give a better understanding of the changes in the underlying dynamics

We strongly agree with the Reviewer that adding mean circulation changes is improving the manuscript which is why we added the change of wind at 850hPa between 2081-2100 compared to the reference period as multi-model mean. We also discuss the most relevant changes and compare individual model projections.

[Figure]

Figure 1. Change in wind vectors (850hPa) and wind speed (m/s) in 2081-2100 compared to the reference period in the MMM of the TOP6 models.

[Figure]

*Figure 2. Change in wind vectors (850hPa) and wind speed (m/s) in 2081-2100 compared to the reference period for the TOP6 models.*

3. The "wet-getter-wet" argument is not new. If you can look at the thermodynamic and dynamic components of the precipitation change, it can give a better insight.

We agree with the reviewer, that better insight is provided if adding an analysis of the thermodynamic and dynamic components. However, Xue et al. (2023) has provided these results in the meantime, which is we decided not to reproduce the same results within this manuscript. However, if the reviewer would like to see these results added, we are happy to add them to the revised manuscript.

*Xue, D., Lu, J., Leung, L.R. et al. Robust projection of East Asian summer monsoon rainfall based on dynamical modes of variability. Nat Commun 14, 3856 (2023). https://doi.org/10.1038/s41467-023-39460-y*

---

## Author Comment (AC2)

We thank the reviewers for the valuable comments that contributed to a substantially improved version of the manuscript. The major changes resulting from the two reviewer reports obtained are:

- adding further model evaluation criteria (interannual variability, spatial distribution, 850hPa wind) resulting in some changes in the selected model set

- focusing the results on the best performing models only, called TOP6 throughout the manuscript and adapting previous results accordingly

- focusing on area that additionally fulfills monsoon definition (JJA minus DJF rainfall exceeds 2 mm/day) and providing results only based on this area

- using observational data (GPCC) instead of reanalysis data (W5E5) for reference

- discussing underlying physical mechanisms for changes in mean circulation (wind 850 hPa)

- adding a significance measure to give insight into the robustness of the projections following IPCC standards

- adding new subchapter regarding the wet bulb temperature projections

We are looking forward to further feedback from the reviewers.

**-----------------------**

Reviewer #2

**-----------------------**

General comment:

This study examined future changes of East Asian Summer Monsoon (EASM) including seasonal mean precipitation, interannual variability and extreme wet seasons using CMIP6 models. According to the simulated magnitude of regional, seasonal mean precipitation, the CMIP6 models are divided into two groups. The projections are then based on the group of models with reasonable magnitude of EASM precipitation. It is shown that mean precipitation, interannual variability and extreme wet seasons of EASM will increase in the future under different SSP scenarios.

However, the current analyses lack significance and robustness in a few aspects.

Firstly, many methods are inappropriate.

(1) The model evaluation is based on regional mean and seasonal mean precipitation over East Asia. But the EASM is a complex system, in which the monsoon circulation and spatial pattern of precipitation is very important. Hence this evaluation metric is insufficient to represent the EASM, which means that the model selection is not robust.

We added a broader set of model selection criteria including:

-    The mean JJA rainfall is within two standard deviations of the observed mean in the GPCC dataset (1995-2014).

- The model's standard deviation is within plus/minus 50% of the observed GPCC standard deviation (1965-2014).

- The centered root mean square error (CRMSE) is smaller than 2 mm/day (1995-2014).

- The main features of the EASM circulation (southwest winds originated from the Bay of Bengal and western flank of the tropical Western Pacific High) are captured according to the JRA-55 dynamics (1995-2014)

In the revised manuscript, we performed the model selection on basis of these criteria resulting in 6 models with better performance regarding the EASM that we call TOP6. We provide new figures and a new table to present the results. Besides, we adapt all results throughout the paper accordingly.

[Figure]

*Figure 1. Wind vectors at 850hPa and wind speed (m/s) for 1995-2014 for the CMIP6 models with best performance regarding EASM (TOP6).*

[Figure]

*Figure 2. Spatial distribution of EASM averaged over the period 1995-2014 from the TOP6 CMIP6 models*

(2) Reanalysis, rather than observational data, is used to evaluate the model simulated precipitation. As there are many observational datasets (e.g., APHRODITE, GPCC, GPCP) in East Asia, they should be used for model evaluation.

In the revised manuscript, we use GPCC as observational reference data.

[Figure]

*Figure 3. Spatial distribution of EASM averaged over the period 1995-2014 (GPCC data).*

(3) As the focus is the EASM, the EASM domain could be defined more appropriately, taking into account many proposed definitions in previous studies.

We adapted now the definition for monsoon domain following the IPCC AR6 (JJA minus DJF mean rainfall exceeding a threshold of 2mm/day).

[Figure]

*Figure 4. East Asian summer monsoon area within 20-50°N and 100-150°E as covered in the revised manuscript.*

(4) Significance test for projected changes needs to be included. These limitations in methods weaken the robustness of the results.

We added significance checks following the IPCC AR6: The signal is classified as robust, where >=66% of models show change greater than the variability threshold and >=80% of all models agree on sign of change.

Secondly, no physical understanding is provided at all. The study focuses on model simulated change in regional mean EASM precipitation, but no physical understanding of EASM change is provided.

We added a discussion regarding the changes in the mean circulation and the underlying physical dynamics. We also provide the MME changes of TOP6 model in 850hPa wind.

[Figure]

*Figure 5. Changes in wind vectors at 850hPa and wind speed (m/s) for 1995-2014 for the CMIP6 models with best performance regarding EASM (TOP6 Multi-model mean).*

Most importantly, the scientific question needs to be refined in order to improve our current knowledge about EASM change. There have been plenty of papers investigating the future changes of EASM covering a wide variety of aspects, including process-based projection, spatial pattern of precipitation change, thermodynamic and dynamic processes of monsoon precipitation change, changes in monsoon duration, northern boundary, extremes (see a few references below and references therein). For example, based on CMIP6 multi-models, there is dynamic-based projection of EASM rainfall and variability (Xue et al., 2023), projection of monsoon rainfall and duration with thermodynamic and dynamic understanding (Moon and Ha, 2020; Ha et al., 2020). Overall, these studies so far have provided a relatively robust understanding of future EASM change based on GCMs. The current study, unfortunately, does not add improvements in our current knowledge. Thus, the authors need to evaluate carefully the current knowledge gap regarding EASM change and refine the scientific questions to be investigated.

References:

Xue, D., Lu, J., Leung, L.R. et al. Robust projection of East Asian summer monsoon rainfall based on dynamical modes of variability. Nat Commun 14, 3856 (2023). https://doi.org/10.1038/s41467-023-39460-y

Moon, S., Ha, KJ. Future changes in monsoon duration and precipitation using CMIP6. npj Clim Atmos Sci 3, 45 (2020). https://doi.org/10.1038/s41612-020-00151-w

Ha, K.-J., Moon, S., Timmermann, A.,& Kim, D. (2020). Future changes of summer monsoon characteristics and evaporative demand over Asia in CMIP6 simulations. Geophysical Research Letters,47, e2020GL087492.https://doi.org/10.10

To our knowledge, none of the listed or other existing studies provides insights regarding the interannual variability and associated extremely wet seasons (while extremes on other scales have been discussed). Besides, no other study provides the projections for the monsoon region over China specifically, which is of high relevance for policy makers. Besides, most of these studies regarding future projections lack a model selection process that is tailored for the specific EASM region.

However, we are delighted to add a further section regarding the wet bulb temperature projections over the region in the revised manuscript.

---

## Author Response (AR1)

We thank the two anonymous reviewers for the valuable comments that contributed to a substantially improved version of the manuscript. The major changes resulting from the two reviewer reports obtained are:

- adding further model evaluation criteria (regarding interannual variability, spatial distribution, 850hPa wind) resulting in some changes in the selected model set

- focusing the results on the best performing models only, called TOP6 throughout the manuscript and adapting previous results accordingly

- focusing on area that additionally fulfills monsoon definition (JJA minus DJF rainfall exceeds 2 mm/day) and providing results only based on this area

- using observational data (GPCC) instead of reanalysis data (W5E5) for reference

- discussing underlying physical mechanisms for changes in mean circulation (by analysing wind 850 hPa) and adding a new subchapter regarding changes in circulation that reveal a northward shift of the ITCZ that is responsible for the increase in rainfall in South China.

- adding a significance measure to give insight into the robustness of the projections following IPCC standards

We are looking forward to the feedback from the reviewers.

**-----------------------**

Reviewer #1

**-----------------------**

Review of "Consistent increase of East Asian Summer Monsoon rainfall and its variability under climate change over China in 34 coupled climate models" by Anja Katzenberger and Anders Levermann

General Comments: In this paper, the authors examined the future changes in the mean precipitation over the EASM region as well as its variability under different emission scenarios. Their analysis suggests that both mean precipitation and its variability are increasing under all emission scenarios, with a stronger response under stronger emission scenarios. There were many studies using CMIP5/6 model simulations that looked at the precipitation changes over different monsoon regions of the globe. Also, they find that the "wet-gets-wetter" arguments holds true for EASM region. Overall, the types of analyses presented in this paper are useful for regional climate change assessments. I have some specific concerns that need to be addressed before accepting the article.

Specific comments:

1. My major concern in this paper is the way in which the models are grouped. I don't have any issues with the "Group A" models which have a mean precipitation withing +/- 2 std of the observations (reanalysis). However, the Group B consists of the models in both sides of the extremes. This means that when you take the ensemble mean, you are averaging the outliers on two sides and as a result there may not be any

use of grouping the models in this way. You may either focus on the Group A models or have three groups (one group each for outliers on each side).

We agree with the reviewer that focusing on group B models as an individual group does not provide very useful insights. Thus, we follow the reviewer's proposition to focus on these best performing models. In the revised manuscript, we also refined the selection of Group A models by adding further selection criteria (STD, CRSME, WIND 850 hPa) resulting in an adapted group of selected models, called TOP6 (lines 90f and 106f).

We also want to briefly note that it can be useful to show the model results for a multi-model mean of all models (not group B models only), particularly because in this case the overestimating models compensate for the underestimating models and the multi-model mean results in often showing good performance. See e.g.

*Sing & AchutaRao (2018): Quantifying uncertainty in twenty-first century climate change over India*

*Tebaldi, C. & Knutti, R (2007): The use of multi-model ensemble in probabilistic climate projections*

2. It would be interesting to see the seasonal mean circulation changes as well. This will give a better understanding of the changes in the underlying dynamics

We strongly agree with the Reviewer that adding mean circulation changes is improving the manuscript which is why we added the change of wind at 850hPa between 2081-2100 compared to the reference period as multi-model mean. We also discuss the most relevant changes and compare individual model projections (lines 160f):

*The TOP6 multi-model mean projects that the northeastward winds over the Bay of Bengal in 0-10° N will weaken by up to 3m/s, while they will intensify in 0-20° N. This indicates a northward shift of these southwest winds and strengthens the moisture supply to South China where an increase in rainfall is projected by 5 out of 6 models. This shift in wind patterns is associated with a northward shift of the ITCZ originated in the warming land temperatures due to climate change. The most intense wind change is projected by EC-Earth3 and IPSL-CM6A-LR and the only model that does not project this trend is MRI-ESM2-0.*

*Additionally, half of the TOP6 models (EC-Earth3, GFDL-CM4, MPI-ESM1-2-LR) project that the southwinds originated in the South China Sea will have an increasing tendency towards east. However, this is not a robust finding given the strong intermodel spread in this region.*

[Figure]

*Figure 1. Change in wind vectors (850hPa) and wind speed (m/s) in 2081-2100 (SSP5-8.5) compared to the reference period in the MMM of the TOP6 models.*

[Figure]

*Figure 2. Change in wind vectors (850hPa) and wind speed (m/s) in 2081-2100 compared to the reference period for the TOP6 models.*

3. The "wet-getter-wet" argument is not new. If you can look at the thermodynamic and dynamic components of the precipitation change, it can give a better insight.

We agree with the reviewer, that better insight is provided when adding thermodynamic and dynamic components. However, another publication has provided these results in the meantime, which is we decided not to reproduce the same results within this manuscript. We cite this study with regard to this contribution in line 207/208.

Xue, D., Lu, J., Leung, L.R. et al. Robust projection of East Asian summer monsoon rainfall based on dynamical modes of variability. Nat Commun 14, 3856 (2023). https://doi.org/10.1038/s41467-023-39460-y

**----------------------**

Reviewer #2

**----------------------**

General comment:

This study examined future changes of East Asian Summer Monsoon (EASM) including seasonal mean precipitation, interannual variability and extreme wet seasons using CMIP6 models. According to the simulated magnitude of regional, seasonal mean precipitation, the CMIP6 models are divided into two groups. The projections are then based on the group of models with reasonable magnitude of EASM precipitation. It is shown that mean precipitation, interannual variability and extreme wet seasons of EASM will increase in the future under different SSP scenarios.

However, the current analyses lack significance and robustness in a few aspects.

Firstly, many methods are inappropriate.

(1) The model evaluation is based on regional mean and seasonal mean precipitation over East Asia. But the EASM is a complex system, in which the monsoon circulation and spatial pattern of precipitation is very important. Hence this evaluation metric is insufficient to represent the EASM, which means that the model selection is not robust.

We added a broader set of model selection criteria (see lines 90f) including:

- The mean JJA rainfall is within two standard deviations of the observed mean in the GPCC dataset (1995-2014).

- The model's standard deviation is within plus/minus 50% of the observed GPCC standard deviation (1965-2014).

- The centered root mean square error (CRMSE) is smaller than 2 mm/day (1995-2014).

- The main features of the EASM circulation (southwest winds originated from the Bay of Bengal and western flank of the tropical Western Pacific High) are captured according to the JRA-55 dynamics (1995-2014)

In the revised manuscript, we performed the model selection on basis of these criteria resulting in 6 models with better performance regarding the EASM that we call TOP6. We provide new figures and a new table to present the results. Besides, we adapt all results throughout the paper accordingly.

Fig. 1 & 2 give a first insight into the refinement of the model selection criteria. All the details can be found in lines 90f and 105f.

[Figure]

*Figure 3. Wind vectors at 850hPa and wind speed (m/s) for 1995-2014 for the CMIP6 models with best performance regarding EASM (TOP6).*

[Figure]

*Figure 4. Spatial distribution of EASM averaged over the period 1995-2014 from the TOP6 CMIP6 models*

(2) Reanalysis, rather than observational data, is used to evaluate the model simulated precipitation. As there are many observational datasets (e.g., APHRODITE, GPCC, GPCP) in East Asia, they should be used for model evaluation.

In the revised manuscript, we use data from the Global Precipitation Climatology Centre (GPCC) as observational reference data, see Fig. 3.

[Figure]

Figure 5. Spatial distribution of EASM averaged over the period 1995-2014 (GPCC data).

(3) As the focus is the EASM, the EASM domain could be defined more appropriately, taking into account many proposed definitions in previous studies.

We adapted now the definition for monsoon domain following the IPCC AR6 (JJA minus DJF mean rainfall exceeding a threshold of 2mm/day). Fig. 4 shows the adapted region.

[Figure]

Figure 6. East Asian summer monsoon area within 20-50°N and 100-150°E as covered in the revised manuscript.

(4) Significance test for projected changes needs to be included. These limitations in methods weaken the robustness of the results.

We added significance checks following the IPCC AR6: The signal is classified as robust, where >=66% of models show change greater than the variability threshold and >=80% of all models agree on sign of change. See also lines 143-145.

Secondly, no physical understanding is provided at all. The study focuses on model simulated change in regional mean EASM precipitation, but no physical understanding of EASM change is provided.

We added a subsection that focus on the question: What are the underlying physical mechanism for the projected change in monsoon rainfall? We find that there is a northward shift of the southwest winds that is associated with a northward shift of the ITCZ. We also provide the MME changes of TOP6 model in 850hPa wind speed and direction, see Fig. 5. Changes are in lines 160f of the revised manuscript.

*The TOP6 multi-model mean projects that the northeastward winds over the Bay of Bengal in 0-10° N will weaken by up to 3m/s, while they will intensify in 0-20° N. This indicates a northward shift of these southwest winds and strengthens the moisture supply to South China where an increase in rainfall is projected by 5 out of 6 models. This shift in wind patterns is associated with a northward shift of the ITCZ originated in the warming land temperatures due to climate change. The most intense wind change is projected by EC-Earth3 and IPSL-CM6A-LR and the only model that does not project this trend is MRI-ESM2-0.*

*Additionally, half of the TOP6 models (EC-Earth3, GFDL-CM4,MPI-ESM1-2-LR) project that the southwinds originated in the South China Sea will have an increasing tendency towards east. However, this is not a robust finding given the strong intermodel spread in this region.*

[Figure]

*Figure 7. Change in wind vectors (850hPa) and wind speed (m/s) in 2081-2100 (SSP5-8.5) compared to the reference period in the MMM of the TOP6 models.*

Most importantly, the scientific question needs to be refined in order to improve our current knowledge about EASM change. There have been plenty of papers investigating the future changes of EASM covering a wide variety of aspects, including process-based projection, spatial pattern of precipitation change, thermodynamic and dynamic processes of monsoon precipitation change, changes in monsoon duration, northern boundary, extremes (see a few references below and references therein). For example, based on CMIP6 multi-models, there is dynamic-based projection of EASM rainfall and variability (Xue et al., 2023), projection of monsoon rainfall and duration with thermodynamic and dynamic understanding (Moon and Ha, 2020; Ha et al., 2020). Overall, these studies so far have provided a relatively robust understanding of future EASM change based on GCMs. The current study, unfortunately, does not add improvements in our current knowledge. Thus, the authors need to evaluate carefully the current knowledge gap regarding EASM change and refine the scientific questions to be investigated.

References:

Xue, D., Lu, J., Leung, L.R. et al. Robust projection of East Asian summer monsoon rainfall based on dynamical modes of variability. Nat Commun 14, 3856 (2023). https://doi.org/10.1038/s41467-023-39460-y

Moon, S., Ha, KJ. Future changes in monsoon duration and precipitation using CMIP6. npj Clim Atmos Sci 3, 45 (2020). https://doi.org/10.1038/s41612-020-00151-w

Ha, K.-J., Moon, S., Timmermann, A.,& Kim, D. (2020). Future changes of summer monsoon characteristics and evaporative demand over Asia in CMIP6 simulations. Geophysical Research Letters,47, e2020GL087492.https://doi.org/10.10

We carefully rechecked the existing literature and came to the conclusion that a comprehensive study including the presented characteristics based on models that are performing well in the specific region is still lacking.

Thus, we are happy to summarize and highlight the aspects that we believe to be relevant and sufficient contributions within the scope of Earth System Dynamics.

None of the listed or other existing studies provides insights regarding (1) the interannual variability and (2) associated extremely wet seasons that have been associated with impactful widespread flooding events (e.g. Volonté et al. 2021; while extremes on other scales have indeed been discussed)

Besides, almost all of the listed studies regarding future projections lack a model selection process that is tailored for the specific EASM region – which is, as the reviewer pointed out correctly when requesting more detailed model selection criteria in the context of this review, a very particular monsoon system with very individual regional conditions and thus requires individual attention. Thus, we think it is also important to (3) provide the mean projections (and other results) based on models with regional satisfying performance in the past climate.

Besides, (4) we added a new subsection that discusses changes in circulation patterns. This includes the finding that the intensification over South East China is associated with a northward shift of the ITCZ.

Additionally, no other study provides (5) the projections for the monsoon region over China specifically, which is of high relevance for policy makers.

We hope that these highlights could convince the reviewer of the relevance of our contribution.

---

## Author Response (AR2)

**Editor Comment**

Dear Anja and Anders,

Sorry for the long time it took me to make a decision about your paper. I went personally through your manuscript and read the replies to reviewers. I am mostly satisfy and ready to publish it. I have only one remark that I think you should address in your paper.
EASM is a different kind of monsoon than tropical ones that are linked to ITCZ dynamics. According to "Zhisheng, A., Guoxiong, W., Jianping, L., Youbin, S., Yimin, L., Weijian, Z., ... & Juan, F. (2015). Global monsoon dynamics and climate change. Annual review of earth and planetary sciences, 43, 29-77" EASM is a part of subtropical monsoons, which dynamics is more related to interactions between large-scale topography, the Rossby radius of deformation and the jet stream (Molnar et al. 2010). How this would fit with your conclusions, namely "The rainfall increase in South-East china is due to a northward shift of the southwest winds associated with a northward shift of the ITCZ that strengthens the water supply towards this region"? Additionally, simply pointing to Xue et al., 2023 regarding thermodynamics vs dynamics at lines 241-242 is not enough in my view: you have still space in conclusions to expand a bit on mechanisms.

Small typo at ln 23: May with capital M.

All the best,
Roberta D'Agostino

**Answer to Editor**

Dear Dr. Roberta D'Agostino,

We thank you for personally going to all reviewer comments and our response and are very delighted to hear that you are almost satisfied with the revised manuscript. We also very much appreciate the remaining remark about the dynamics and reviewed further literature as well as discussed this aspect in detail. As a result, we came to the conclusion that the additional moisture supply associated with the ITCZ northward shift as proposed in the earlier version of our manuscript is a minor dynamic effect that is not sufficient to explain the projected EASM rainfall increase. Rather, the lack of circulation changes in the study region itself underlines a minor role of the dynamic component and points to the dominating role of the thermodynamic component instead. This is also in line with the findings of existing studies (e.g. Li et al. 2015; Lee et al. 2017; Li et al. 2021). We adapted the manuscript accordingly. We also expanded the discussion about the components as requested and corrected minor typos. You can find the updated conclusion on the next page or in the revised manuscript. Thanks to the Editor's valuable comments, we now believe to submit a significantly improved version of the manuscript and are looking forward to hearing from the editor.

All the best,
Anja

*"The multi-model mean of the wind pattern reveals relatively minor changes in the circulation in the region. This indicates that there are only small changes in the dynamic component within East Asia pointing to the dominant contribution from the thermodynamic component. Indeed, this is in line with the CMIP6 study of Li et al. (2021) who quantified the role of the different components contributing to the EASM increase throughout the 21st century. In East China (Japan and Korea Region) long-term, they quantify the change in moisture advection to be +9.6% (+9.2%), evaporation +19.9 % (+16.1%) and moisture convergence +70.6% (+74.4%). Additionally, they split the moisture convergence term into a term that relates to circulation changes (dynamic changes), one that refers to moisture content changes (thermodynamic changes) and a residual term that can be assumed to be small. In East China (Japan and Korea Region) long-term, the thermodynamic term clearly dominates with +98.1% (+153.0%) over the dynamic term of +3.0% (-34.9%). The authors find that the dynamic term might even be cancelled out due to the large intermodel spread (Li et al., 2021). This intermodel spread might also at least partly explain that the dynamic component has been found to contribute positively as well as negatively to the budget (Wang et al., 2014; Li et al., 2015; Lee et al., 2017; Li et al., 2021). However, most studies coincide with the dominant thermodynamic role in the region (Li et al., 2015; Lee et al., 2017; Li et al., 2021). But there is also one study that finds that the dynamic component might be dominating in the region with 67% over the 33% of the thermodynamic component (Xue et al., 2023). However, as pointed out by the authors of this study, the projections are based on a single model (CESM2) that in our study was also not among the best performing regarding the EASM characteristics.*

*The relevance of evaporation and moisture convergence has already been reported in the context of CMIP5 (Seo et al., 2013; Qu et al., 2014). Seo et al. (2013) found that these changes are induced by the (north) westward shift of the North Pacific subtropical high. Along the northern and northwestern flank of the strengthened high, intensified southerly or southwesterly winds lead to an increase in moisture convergence, intensifying precipitation particularly over the Baiu region to the east of Japan and the continental region to the north of the Korean Peninsula. Qu et al. (2014) also add the increased vertical transport of moisture in the EASM region and the capacity of warmer air to hold more moisture following Clausius-Clapeyron as relevant contributing factors. In other studies, the role of the strengthening of the land-sea thermal contrast under global warming is discussed (Endo et al., 2018)."*